# A Mediation Model of Self-Efficacy and Depression between Burnout and Alcohol Consumption among Health Workers during the COVID-19 Pandemic

**Alejandra del Carmen Domínguez-Espinosa \***, **Fátima Laborda Sánchez**, **Alma Mireya Polo Velázquez** and **Graciela Polanco Hernández**

Psychology Department, Iberoamerican University, Mexico City 01219, Mexico
* Correspondence: alejandra.dominguez@ibero.mx; Tel.: +52-5559504000 (ext. 4739)

**Abstract:** To verify the role played by burnout in the prediction of alcohol consumption, considering the integration of the theory of job demands-resources (JD-R) and sociocognitive theory (SCT), we developed an analytical model in which self-efficacy and depression act as mediators of this relationship. A cross-sectional online survey was taken by 3856 workers enrolled in various public agencies of the Mexican Health Ministry during the COVID-19 pandemic in Mexico. The results indicate that of the three dimensions of burnout, only depersonalization predicts alcohol consumption; however, self-efficacy regulates the effect of emotional exhaustion and achievement dissatisfaction on alcohol consumption. Similarly, the three components of burnout have indirect effects through depression, suggesting that depression and self-efficacy mediate the relationship between burnout and alcohol consumption. Burnout alone cannot explain alcohol consumption, but when depression is present, burnout increases the predisposition to consume alcohol; when self-efficacy is present, the probability of alcohol consumption decreases.

**Keywords:** burnout; alcohol consumption; depression; self-efficacy; COVID-19





## 1. Introduction

The COVID-19 pandemic disruption brought substantial changes to the ways in which employees work and interact within their work environments. Organizational and individual demands have increased since the early stages of the pandemic, and a rise in burnout scores has been widely reported in various scenarios across the health work sector [1–6].

A clear relationship between job demands and health workers' deterioration was established long before COVID-19 [7]. The theory of job demands-resources [8,9] shows that the physical, psychological, social and organizational job demands needed to perform work do not allow employees to recover from stressors; rather, physical, psychological, social and organization job resources are those aspects of the job that are functional and can reduce job demands and their associated costs [10]. Constant exposure to job demands (i.e., stressors) causes energy drain and can result in burnout [11] or other emotional problems, such as anxiety and depression [12–14], culminating in the potential use of substances as a maladaptive coping strategy [5,15] and eventually resulting in adverse work-related outcomes [16]. When considering job resources, most studies include autonomy as the preferred variable [11]. However, another variable coming from sociocognitive theory [17,18], self-efficacy, has proven to be effective when dealing with stress across work environments [19].

To better understand the relationship among work demands, individual resources and their consequences at the emotional and behavioral levels, an analytical model was established in which the COVID-19 pandemic (generalized demand) put health workers under immense stress, leading them to experience burnout (personal demand), with a

consequent increase in the probability of consuming alcohol (personal outcome); self-efficacy and depression (a personal resource and a personal lack of resources, respectively) act as mediators of this relationship.

## 1.1. Job Demands and Resources Model

The COVID-19 pandemic is considered to be one of the main sources of extreme stress in the last decade. It resulted in work overload in all health work sectors [20]. Examples of frequent stressors that affect health workers (e.g., doctors, nurses and pharmacists) include the need to be on call for 24-hour periods, sleep deprivation, an increase in the number of omissions and errors [21,22], scarce material and work equipment (e.g., protective equipment) and the risk of getting sick while working [2,23–26]. During the COVID-19 pandemic, additional stressors emerged: grief over the death of friends or loved ones due to infection [27,28], fear of infections, the fatality rate among health workers [29,30] and dealing with social stigma and discrimination [31,32]. With all of these stressors (i.e., job demands), health workers were exposed to worse conditions in their work environments, which increased their chances of experiencing burnout [33].

Burnout, which was originally described by Freudenberger [34] and then by Maslach [35,36], is a process that includes three dimensions: (1) emotional exhaustion, which refers to the drain of emotional resources due to job demands; (2) depersonalization, the interpersonal component of defensive coping characterized by cynicism; (3) reduced personal accomplishment or achievement dissatisfaction, the self-evaluation component characterized by a sense of inefficacy and failure [37,38]. This process is the result of a prolonged response to stressors without being able to recover from them in due time [39]. Burnout is usually considered to be a part of the global strain on a person's well-being; this strain usually involves anxiety and depression.

The job demands-resources (JD-R) model [8,9] indicates that resources can ameliorate the consequences of stressors that eventually cause burnout and distress [10,40]. Job resources are factors that facilitate the achievement of work-related goals, help reduce job demands and increase work engagement [11]. Job resources such as adequate rest, social support, access to protective equipment and a satisfactory work environment can reduce burnout and other psychological problems [41,42].

## 1.2. Alcohol Consumption

Most health workers can cope with stress in normal conditions, but the COVID-19 pandemic has placed excessive demands on health workers, affecting their sense of efficacy in dealing with their job demands [43] and consequently diminishing their emotional states.

The relationship between burnout and alcohol consumption has been explored with mixed results [44–46]; some studies did not find a significant association between burnout and alcohol consumption or found an association only when another variable was present [47–50]. According to Frone [51], only those workers who cannot cope with stressors or have certain vulnerabilities use alcohol as a way of dealing with stress. During the COVID-19 pandemic, the consumption of alcohol increased due to different job stressors, including job loss, which resulted in higher physical and mental risk, thereby leading to anxiety, depression and feelings of loneliness [52–57].

Specifically, people with records of alcoholism who were infected with COVID-19 showed higher liver dysfunctionality than those who were not infected [58]. Additionally, it was reported that hepatic alterations ranging from mild to severe were observed from infection with SARS-CoV-2 [59,60], and an increased risk of severe COVID-19 infection and admission to the ICU [61] was observed among patients living with nonalcoholic fatty liver disease (NAFLD), just to mention some examples.

It was proposed that burnout and alcohol consumption are moderated by job control [62]; nevertheless, the direct and indirect effects were limited, and in terms of longitudinal consequences, the results proved to be nonsignificant. Therefore, there are still

concerns about the relationship between job demand and job resources and their link to alcohol consumption.

The JD-R model also implies that individual resources can help regulate the impact of stressors on mental health. Self-efficacy is a set of self-beliefs that can enhance or hinder motivation; it is one's perceived capability to produce a given level of achievement [63,64]. Self-efficacy is an important mechanism of human self-regulation that supports workers in maintaining mental health and coping with stress [65]. Self-efficacy beliefs about managing negative emotions at work significantly mediate the longitudinal relationship between emotional stability and burnout [66]. More specifically, self-efficacy has been reported to help reduce burnout [2,41,65], which is extensive in various professions and work contexts [67,68].

Therefore, the purpose of the present study was to test an analytical model in which the COVID-19 pandemic (generalized demand) has put health workers under immense stress and led them to experience burnout (personal demand), with a consequent increase in the probability of consuming alcohol (personal outcome), where self-efficacy and depression (a personal resource and a personal lack of resources, respectively) mediate the relationship.

## 2. Materials and Methods

We conducted data collection during the 2nd and 3rd COVID-19 waves in Mexico as a part of a psychoeducational intervention organized by the Mexico City Health Ministry and the Labor Union of the Mexican Social Security Institute. The courses and participation in the cross-sectional study were offered to all affiliated workers, including physicians, nurses, administrators, pharmacists, security workers, etc. throughout the COVID-19 pandemic. Enrollment in the intervention program and taking the survey were both voluntary and anonymous processes. At the beginning of the online survey, the participants gave their consent to start responding. Participants needed to be active workers and to give their consent to participate in the study; they also needed to have an active email account to which the invitation to participate in the research could be sent. No exclusion criteria were established, and the only elimination criteria were incomplete data in the sociodemographic section and more than 20% missing data in the complete survey.

### 2.1. Participants

A non-probabilistic sampling strategy was used, and a self-selected sample of health sector workers agreed to participate voluntarily. A total of 4713 workers started the survey, but only 3856 workers finished, representing a loss of 18% of participants from the original self-selected sample. See Supplemental Materials Figure S1.

### 2.2. Instruments

The survey was part of a larger applied study on the mental health of health workers with a pilot intervention program aimed at ameliorating the potential distress suffered by health workers during the COVID-19 pandemic. The survey was conceived as a screening tool for the participants to enroll in a series of online free courses. Participation in the online courses and the screening test were completely voluntary and anonymous, and the participants signed an informed consent form before enrolling.

In the current study, we focused on the responses on four scales included in the survey, which were presented to each participant in the same order. In the sociodemographic section, the participants were asked to indicate their gender, age, marital status, educational level and main area of work. Immediately thereafter, the participants were presented with the following psychological scales in the subsequent order:

2.2.1. Burnout Scale

The Burnout Mexican Scale from Uribe Prado [38] was adapted to be used with heath workers. Twelve items that represent and capture the three-dimensional model of Maslach and Jackson [69] were selected from the original: five items on emotional exhaustion

(e.g., "When I get home from work, all I want is to rest"), three items on depersonalization (e.g., "I have a hard time being polite to patients or other colleagues at work") and four items on achievement dissatisfaction (e.g., "I feel that my skills and knowledge are wasted at work"). Evidence of three-dimensionality was obtained through exploratory factor analysis with principal component extraction and Promax rotation, which yielded 69.63% of the explained variance in total with eigenvalues of 5.68 (47.39%), 1.42 (11.87%) and 1.24 (10.36%); Cronbach's alphas of 0.87, 0.83 and 0.84; McDonald's omega values of 0.88, 0.83 and 0.85, respectively.

### 2.2.2. Depression Scale

The depression scale was derived from the Center for Epidemiological Studies Depression Scale, which is known as CESD-47 in its Spanish version [70]. It involves seven questions (e.g., "Did you feel like you could not shake the sadness off?"), including a general item about depression ("Did you feel depressed?"). The scale explores dysphoric mood, motivation, concentration, loss of pleasure and poor sleep [71]. Evidence of one-dimensionality was obtained from an exploratory factor analysis with principal component extraction and Promax rotation with an explained variance of 58.27%; an eigenvalue of 4.46; a reliability index with a Cronbach's alpha and McDonald's omega of 0.90 and 0.91, respectively.

### 2.2.3. Self-Efficacy Index

The self-efficacy index was inspired by the indicators proposed by Balasubramanian et al. [72] for personal positive measures that can help individuals cope with stressful and demanding situations during the COVID-19 pandemic. Balasubramanian et al. [72] indicated that setting realistic goals and having a checklist and work plan for the day can decrease stress. These indicators were taken and paraphrased to be declarative statements to be answered in the third person and in past tense, as they operationalized how well health professionals and staff were coping with current demands. We developed four items: (1) "I have set realistic goals at work"; (2) "I have felt capable of facing problems at work"; (3) "I feel able to put work problems in their proper dimension"; (4) "I have felt capable of finding a solution to my problems at work". By means of an exploratory factor analysis with principal components and Promax rotation, a single dimension was established with 71.58% of explained variance, an eigenvalue of 2.86 and a reliability index with a Cronbach's alpha and McDonald's omega of 0.86 and 0.87, respectively.

### 2.2.4. Alcohol Consumption

The Alcohol Use Disorders Identification Test, better known as AUDIT, was developed for the World Health Organization in its Spanish version and in a self-report format [73]. It includes ten items, and the scoring scheme weights the frequency by the amount of alcohol consumed. The sum can be used with cutoff points to classify participants into three or four consumption groups. For the present study, the total score was used as a linear variable in the structural equation model. To check for one-dimensionality, we ran an exploratory factor analysis with principal component and Promax rotation, and the total variance achieved was 41.97% with an eigenvalue of 4.19 and a corresponding reliability index with a Cronbach's alpha of 0.81 and a McDonald's omega of 0.83.

### *2.3. Procedure*

The data collection was performed online using Research Electronic Data Capture (REDCap, [74]). The participants were invited to enroll in an intervention program in which a series of short courses were offered as resourceful tools for coping with distress. The courses were offered to the workers through a closed platform. The invitations to participate in the courses were sent via email and announced within intranet mail services. The authors of the manuscript did not have direct access to the email list, and health institution staff members sent the invitations.

Before the course videos were presented, a pop-up window asked the participants whether they would be interested in participating in a survey. If the participant agreed to participate, then an informed consent form was displayed. After providing informed consent, the participants could start the survey and could pause or drop out at any time, and the participants had the option to go directly to the online course material instead. When the participants finalized the survey, they received feedback on the results, and if their scores were high, they were shown an informative note directing them to seek psychological services. The participants were redirected to the online courses after they finished the survey. The survey lasted approximately 25 min.

*2.4. Data Analyses*

A structural equation model (SEM) was proposed (see Figure 1), and the maximum likelihood estimation strategy was used due to its omnibus test. The analyses were performed using IBM SPSS 29 and AMOS 29 [75,76]. All fit indexes were compared to the suggested threshold to indicate an acceptable fit [77,78]. A total of 5000 bootstrap samples were used to estimate 95% confidence intervals (CIs) to address outliers and assess the significance of the mediation effects [79]. Descriptive analysis, zero-order correlations, mean contrasts by gender and working areas, and convergent and discriminant evidence of validity can be found in the Supplemental Materials Tables S1–S5.

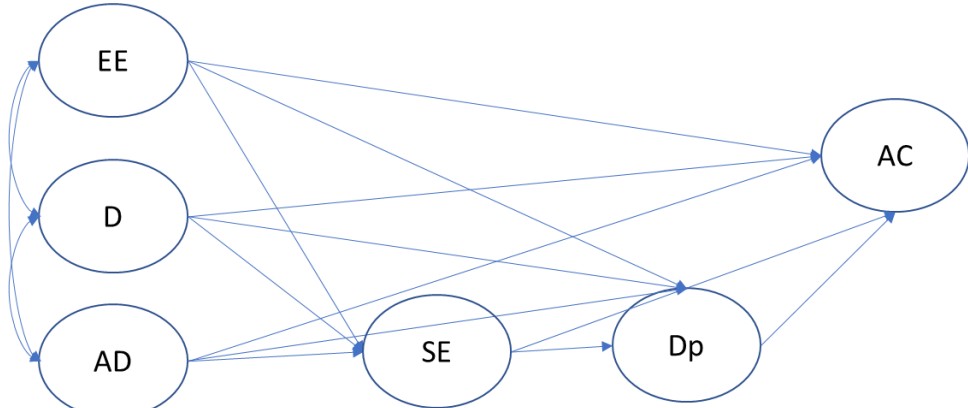

**Figure 1.** Mediation model of self-efficacy and depression between burnout and alcohol consumption. Latent constructs are shown in circles, and observable variables and error terms are not shown; double-headed arrows represent covariances, and single arrows represent direct effects. EE = Emotional Exhaustion; D = Depersonalization; AD = Achievement Dissatisfaction; SE = Self-Efficacy; Dp = Depression; AC = Alcohol Consumption.

## 3. Results

Two hundred and seventy-nine participants were enrolled in clinical and medical care (53.9%), 1081 were working in administrative positions (28%), 69 were collaborating in clinical and laboratory analyses (1.8%) and 627 worked in general services (e.g., security and IT; 16.3%). Of the total sample, 2884 identified themselves as women (74.8%), and 972 identified as men (25.2%), with an age range from 20 to 66 years old (Mage = 38.29 years; S.D. age = 9.5 years); 1892 reported being single or divorced (49.1%), 1964 were married or in a free union (50.9%), 815 had a secondary level of education (years 7 to 12; 21.1%), 1832 had a bachelor's diploma (47.5%) and 1209 had a postgraduate degree (medical specialization, master's and doctoral degrees; 31.4%).

Before testing the model, we checked for normality assumptions and found multivariate outliers, with skew and kurtosis out of range for alcohol consumption. As the purpose of this scale is to estimate extreme consumption, it was expected to have outliers according to epidemiological evidence; therefore, we decided not to delete any outliers and addressed the problem with a bootstrap approach to obtain robust estimates and confi-

dence intervals. The maximum likelihood estimation method yielded the following results: $\chi^2(473) = 2907.918$, $p < 0.001$, CFI = 0.962, TLI = 0.958, RMSEA = 0.037 [0.035–0.038]; these results are considered to indicate good fits and misfits based on suggested thresholds. As shown in Table 1, the three dimensions of burnout had significant negative effects on self-efficacy ($\beta = -0.18$, $\beta = -0.15$, $\beta = -0.12$) and depression ($\beta = 0.54$, $\beta = 0.10$, $\beta = 0.12$), but only depersonalization had a significant positive effect on alcohol consumption ($\beta = 0.15$). Self-efficacy had a significant negative effect on depression ($\beta = -0.11$) and alcohol consumption ($\beta = -0.04$) but to a lesser extent. Depression had a significant positive effect on alcohol consumption ($\beta = 0.18$). Of all the significant estimates, the effect of emotional exhaustion on depression was the largest.

**Table 1.** Structural equation model of burnout predicting alcohol consumption.

| Relationships | B | S.E. | C.R. | β | *p* |
|---|---|---|---|---|---|
| Emotional Exhaustion → | | | | | |
| Self-Efficacy at Work | −0.20 | 0.03 | −7.07 | −0.18 | <0.001 |
| Depression | 0.60 | 0.03 | 24.59 | 0.54 | <0.001 |
| Alcohol Consumption | −0.02 | 0.02 | −1.08 | −0.04 | 0.279 |
| Depersonalization → | | | | | |
| Self-Efficacy at Work | −0.21 | 0.03 | −6.25 | −0.15 | <0.001 |
| Depression | 0.14 | 0.03 | 5.33 | 0.10 | <0.001 |
| Alcohol Consumption | 0.10 | 0.02 | 5.89 | 0.15 | <0.001 |
| Achievement Dissatisfaction → | | | | | |
| Self-Efficacy at Work | −0.19 | 0.04 | −4.86 | −0.12 | <0.001 |
| Depression | 0.18 | 0.03 | 5.79 | 0.12 | <0.001 |
| Alcohol Consumption | 0.00 | 0.02 | 0.21 | 0.01 | 0.835 |
| Self-Efficacy at Work → | | | | | |
| Depression | −0.11 | 0.02 | −7.53 | −0.11 | <0.001 |
| Alcohol Consumption | −0.02 | 0.01 | −2.19 | −0.04 | 0.029 |
| Depression → | | | | | |
| Alcohol Consumption | 0.08 | 0.01 | 6.19 | 0.18 | <0.001 |

Note: Bootstrap sample = 5000 with replacement. $\chi^2$ = 2907.918; GFI = 0.953; TLI = 0.958; CFI = 0.962; RMSEA = 0.037 [.035–.038].

Based on these results, we can partially support the assertion that burnout directly predicts alcohol consumption. Emotional exhaustion and achievement dissatisfaction did not have a direct significant effect on alcohol consumption, and only depersonalization had a significant positive effect on alcohol consumption. Self-efficacy and depression significantly predicted alcohol consumption, and the latter had the greatest effect.

We checked the parallel mediation effects. When the indirect effects of emotional exhaustion on alcohol consumption through depression and self-efficacy were compared, the indirect effects were significant (B = 0.049, B = 0.004), whereas the direct effects were not; therefore, we can conclude that a full mediation effect occurs. The same occurred with achievement dissatisfaction in both paths through depression and self-efficacy on alcohol consumption (B = 0.014, B = 0.004). Taking these indirect effects into account, all mediational analyses showed significant but small effects. The results are displayed in Table 2.

**Table 2.** Indirect effects of burnout and self-efficacy on alcohol consumption.

| | B | Confidence Interval | | *p* |
|---|---|---|---|---|
| | | Low | High | |
| EE -> D -> AC | 0.049 | 0.035 | 0.066 | 0.006 |
| EE -> SE -> AC | 0.004 | 0.001 | 0.007 | 0.030 |
| EE -> SE -> D -> AC | 0.002 | 0.001 | 0.003 | 0.004 |
| DEP -> D -> AC | 0.011 | 0.007 | 0.017 | 0.012 |

**Table 2.** *Cont.*

| | **B** | **Confidence Interval** | | *p* |
|---|---|---|---|---|
| | | **Low** | **High** | |
| DEP -> SE -> AC | 0.004 | 0.002 | 0.009 | 0.014 |
| DEP -> SE -> D -> AC | 0.002 | 0.001 | 0.003 | 0.004 |
| AD -> D -> AC | 0.014 | 0.009 | 0.020 | 0.008 |
| AD -> SE ->AC | 0.004 | 0.001 | 0.009 | 0.013 |
| AD -> SE -> D -> AC | 0.002 | 0.001 | 0.003 | 0.002 |
| SE-> D ->AC | −0.009 | −0.013 | −0.006 | 0.004 |

Note: EE = Emotional Exhaustion, DEP = Depersonalization, AD = Achievement Dissatisfaction, D = Depression, SE = Self-Efficacy; AC = Alcohol Consumption. Unstandardized coefficients reported. Bootstrap sample = 5000 with replacement.

## 4. Discussion

In the present research, we found that only one of the three components of burnout syndrome, depersonalization, positively predicts alcohol consumption, and that emotional exhaustion and achievement dissatisfaction do not, coinciding in part with previous findings that the relation between burnout syndrome and alcohol consumption can be partially attributed to other variables [44–48,50].

In previous research, alcohol consumption and illicit substance use have been shown to be related to depersonalization [46,80], which could be due to the underlying mechanism, explained by Maslach et al. [81], that a lack of sensitivity and a tendency to react in a cold and detached way can be compared with the subjective experience of affective distance and relief response derived from consumption [82]. The negative prediction of alcohol consumption by self-efficacy, based on the previously described association [68], can be identified as a possible protective factor. However, when testing the mediation model, we observed that its protective capacity increased when it impacted depression and subsequently alcohol consumption, which is possibly due to the strong association between depression and alcohol consumption [39,48,83,84] and to the direct effect between these two conditions. In addition, it was found that alcohol consumption is favored when both emotional exhaustion and depression are present.

These results are based on data obtained during the COVID-19 pandemic, during which health personnel reported high levels of loss (real and symbolic) [27,28,85]. We think that the stressful subjective experience of emotional exhaustion matches depressive symptomatology and thereby generates the observed phenomenon. We also think that the increase in alcohol consumption is perhaps an immediate attempt to relieve depressive discomfort.

The negative association between self-efficacy and burnout was confirmed [66,68,86]. Although the components of burnout (emotional exhaustion, depersonalization and achievement dissatisfaction) have a direct negative effect on self-efficacy, self-efficacy negatively predicts depression, leading to decreased levels of alcohol consumption; thus, we can conclude that interventions based on increasing self-efficacy in stressful conditions that cannot be modified, such as the COVID-19 pandemic, are a good alternative for preventing depressive symptoms and consequently reducing alcohol consumption.

Currently, health workers can benefit from the presence of new technologies and procedures for the accurate detection of COVID-19 in all of its variants [87,88]. As already proven in several studies, the association of alcoholism and liver function problems was aggravated by COVID-19 infection [58]. People suffering from illnesses such as NAFLD and alcoholism [57,59,60,89] had increased odds for severe COVID-19 infection and admission to ICUs [61]. For future practice, it is important to monitor the mental and physical status of health workers with a higher likelihood of various predisposing conditions, such as those previously mentioned.

One limitation of the current study is that random sampling procedures were not used for the survey, which limits the generalizability of the findings. All of the participants voluntarily agreed to complete the survey, which may have led to a self-selection bias. The

survey was distributed online, which prevented potential participants with restricted access to electronic infrastructure (e.g., concierges and security guards) from being included.

## 5. Conclusions

The JD-R model was proven to be supported, and the role of self-efficacy was shown to mediate the relationships among burnout, depression and alcohol consumption. In other words, beliefs about one's capacity to deal with stressful situations reduce the probability of experiencing depression and thereby decrease the probability of maladaptive alcohol consumption. In contrast, the probability of falling into maladaptive alcohol consumption is increased by experiencing depression while suffering from burnout. From 2020 to 2022, the consumption of alcohol increased due to old and new stressors associated with the COVID-19 pandemic, which resulted in higher physical and mental risk, leading to anxiety, depression and feelings of loneliness [52–57].

Expanding to other settings, we could take into account economic factors an important stressor during the COVID-19 pandemic. Although that unemployment triggered by the COVID-19 pandemic has recovered in Mexico [90], there are still people who are unemployed; therefore, they are at a higher risk of increasing their dependency to alcohol as a way of coping with economical stress [55]. Regarding health workers, ensuring their salary and benefits would avoid, to some extent, the presence of economical stressors that increase the probability of experiencing depression and problematic alcohol consumption.

The severe effects of burnout, such as depression and increased alcohol consumption, can be prevented by fostering a sense of self-efficacy. Strengthening a sense of competence and reducing institutional stressors (e.g., providing sufficient work materials and relevant and adequate rest and recreation space) will reduce the probability of engaging in unhealthy coping strategies, such as alcohol consumption, among health workers.

Following Kisely et al. [41], we can suggest that authorities at hospital and clinics to implement different strategies to improve health workers mental health, such as: Include short breaks and time off work from as a regular routine; Guarantee effective training; Give positive feedback; Use clear and direct communications with workers; Give opportunity to access to tailored psychological intervention; Guarantee protective and material necessary to work; With these measures, we can foster a sense of self-efficacy and reduce the stressors associated with burnout.

**Supplementary Materials:** The following supporting information can be downloaded at: https://www.mdpi.com/article/10.3390/covid3040046/s1, Table S1. Descriptive statistics and reliability indexes; Table S2. Pearson's correlations across variables; Table S3. Convergent and discriminant evidence of validity; Table S4. Gender difference across composite variables; Table S5. Working-area difference across composite variables; Figure S1. Participant selection flow chart.

**Author Contributions:** Conceptualization, A.d.C.D.-E., F.L.S., A.M.P.V. and G.P.H.; data curation, A.d.C.D.-E.; formal analysis, A.d.C.D.-E.; writing—original draft, A.d.C.D.-E. and F.L.S. All authors have read and agreed to the published version of the manuscript.

**Funding:** This research received no external funding.

**Institutional Review Board Statement:** The review of the ethical aspects of the screening test were supervised by the researchers at the Iberoamerican University and the authorities of the Mexico City Ministry of Health and the Mexican Social Security Institute Labor Union in strict adherence to the Helsinki Declaration and the Regulation of the General Health Law on Research for Health for Mexico.

**Informed Consent Statement:** Informed consent was obtained from all subjects involved in the study.

**Data Availability Statement:** The data can be downloaded at https://doi.org/10.6084/m9.figshare.21180730.

**Acknowledgments:** We thank the staff from the Direction for Training and Self-actualization and Research of the Secretary of Health of Mexico City and the Secretary of Women's Action of the National Union of Social Security Workers for helping to collect the data. We also thank the Continuing

Education Office from the Iberoamerican University for their support in launching the survey on the online platform. We also would like to thank Tonathiu Salcedo for helping upload the questionnaire to the online platform.

**Conflicts of Interest:** The authors declare no conflict of interest.

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
