# Peer review of "A Mediation Model of Self-Efficacy and Depression between Burnout and Alcohol Consumption among Health Workers during the COVID-19 Pandemic"

_covid, doi:10.3390/covid3040046_

Round 1

Reviewer 1 Report

Paper is well written but can be improve by incorporating the following:

1.      List some name of public agencies of Mexican Health Ministry involved in cross-sectional online survey during the COVID-19 pandemic.

2.      Explain about which three dimensions of burnout is speaking about?

3.      Introduction start with COVID-19 pandemic which must have some citatical evidence like: (i) Automated deep transfer learning-based approach for detection of COVID-19 infection in chest X-rays. (ii) Automatic Diagnosis of Covid-19 Related Pneumonia from CXR and CT-Scan Images

4.      Keep single format style for all citation. Check line 27 and 30 and so on.

5.      Line number 30, it is difficult to differentiate between abbreviation and citation: “the theory of job demands-resources (JD-R, 8, 9) shows”.

6.       As you know, Recently so many company like Google, Microsoft, Twitter etc. layoff so many employees which were hired during covid time so Justify the statement – “The COVID-19 pandemic can be considered to increase job demands, as it resulted in work overload in all health work sectors”.

7.      Line number 71, “Job resources are factors that facilitate the achievement of work-related goals, help reduce job demands, and increase work engagement (11)”, in my view along with resources work environment must have also to be consider here.

8.      Some background details about many researchers proposed classification of Covid-19, various types of Pneumonia, Tuberculosis and normal images must be included to give suitable background to the research (i) This can be found in Novel deep transfer learning model for COVID-19 patient detection using X-ray chest images (ii) Metaheuristic-based deep COVID-19 screening model from chest X-ray images

9.      Page 2 line 85: consequently diminishing the already deteriorated emotional state of health workers. Justify.

10.  Is it True that the people lost their job during covid and due to which comes under stress and to avoid this stress they used to alcohol. Yes/No justify.

11.   Line 158: “This strategy required we develop four items:” require updation.

12.  Pg no 4 line 178 Participation in the online courses and the screening test were completely voluntary and anonymous, and the participants signed an informed consent form before enrollment. How we can track their appropriateness while filling the survey means they are giving random feedback or proper feedback?

13.   Discussion must be explained pointwise. Running text become difficult to link with other line.

14.  While doing conclusion JD-R model and other results with some numerical value must be explained here.

15.  Can the author include some mythos about COVID-19 in one or two lines somewhere in the paper

16.  Below reference can improve the introduction section impressively: A novel deep convolutional neural network for diagnosis of skin disease. Traitement du Signal, Vol. 39, No. 5, pp. 1873-1877. https://doi.org/10.18280/ts.390548...Efficient automated disease diagnosis using machine learning models

17.  All references must be in same format. Check for 71 and 73 with others.

Reviewer 2 Report

Thank you for the opportunity to review this manuscript which examines a mediation model explaining the role of depression and self-efficacy o the relationship between burnout and alcohol consumption in a large sample of healthcare workers during the COVID-19 pandemic. The issue is relevant as it is important to gain a nuanced understanding of the pandemic impact on the wellbeing this professional category especially in country contexts like Mexico that are not represented in current literature. The paper is wellwritten, methods are clearly described and results clearly presented. I have a number of suggestions which I believe will enhance the quality of the present manuscript.

 1.  Section 2.3 Procedure should be moved right after section 2.1 Participants and before the section on measures.

 2.  In the Data Analysis section authors should indicate the statistical package softwares used for performing the analysis.

 3.  I suggest authors to improve visibility of Figure 1. Currently many of the labbles results impossible to read do to overlaps with other visual effects.

 4. I wander whether authors checked for gender and professional category differences in terms of measures of depression, burnout and self-esteem, if so these data should be reported. Consider for instance relevant research in this area (https://doi.org/10.1177/09697330221114329; doi: 10.1080/20008198.2021.1968141)

Reviewer 3 Report

The manuscript covers an interesting topic, however, some aspects should be considered before publication.

The introduction is hard to follow. I would suggest strictly focusing on the main aspects, and being synthetic as much as possible. Some information provided in the introduction could be moved to the discussion section.

More information about survey administration is needed.

Section 2.1 participants should be moved to the results section. In the methods, the authors should describe inclusion/exclusion criteria instead of describing the resulting sample. 

Sample size calculation is missing. please add.

Sampling methods are not described.

The flow chart of inclusion/exclusion participants from the final analysis should be added.

in the method section, a paragraph describing the type of variables and how they have been treated is missing. please add.

results: descriptive characteristic of the sample is missing. please add it to the table.

the layout of Table 1 is not very clear. please, find another way to show results.

the id number of ethical approval is missing. please add.

The public health impact of your results is missing. Please add. 

Please, also consider adding implications in practice and policies.

English should be revised. Some sentences are hard to read

Round 2

Reviewer 1 Report

Authors have resolve all the issues so my decision is to accept this paper.

Reviewer 3 Report

I appreciate the efforts performed by the Authors for meeting my suggestions.

However, the flow chart is missing. For the flow chart, I mean to show the number of participants excluded at each step. If exclusion has been made only based on missing data, then specify how many participants were excluded because of that.